# Interaction between Illite and a *Pseudomonas stutzeri*-Heavy Oil Biodegradation Complex

**DOI:** 10.3390/microorganisms11020330

**Published:** 2023-01-28

**Authors:** Lei Li, Yun Yang Wan, Hong Mei Mu, Sheng Bao Shi, Jian Fa Chen

**Affiliations:** State Key Laboratory of Petroleum Resources and Prospecting, Research Centre for Geomicrobial Resources and Application, Unconventional Petroleum Research Institute, College of Geosciences, China University of Petroleum, Beijing 102249, China

**Keywords:** biodegradation, illite, *Pseudomonas stutzeri*, heavy oil, inhibition, stimulation

## Abstract

Illite is a widely distributed clay mineral with huge reserves in Earth’s crust, but its effect on heavy oil biodegradation is rarely reported. This study made an investigation of the interactions between illite and a *Pseudomonas stutzeri*-heavy oil complex (*Pst*HO). Results showed that, although illite exerted a negative effect on *P. stutzeri* degrading heavy oil by inhibiting the biodegradation of 64 saturated hydrocarbons (SHs) and 50 aromatic hydrocarbons (AHs), it selectively stimulated the biodegradation of 45 AHs with a specific structure, and its biogenic kaolinization at room temperature (35 °C) and pressure (1 atm) was observed in *Pst*HO for the first time. The finding points out for the first time that, in *Pst*HO, illite may change the quasi-sequential of AHs biodegradation of heavy oil, as well as its kaolinization without clay intermediate.

## 1. Introduction

Illite clay minerals account for more than half of the clay minerals in Earth’s crust [1]. However, the majority of the studies on the effects of clay minerals on crude oil biodegradation by microorganisms have mainly focused on montmorillonite (Mon; for all abbreviations in this paper, see Table 1), kaolinite (Kao), saponite (Sap), palygorskite (Pal), vermiculite, and nontronite (Table 2). To the best of our knowledge, possibly because illite does not possess a remarkable specific surface area (SSA) [2], cation exchange capacity (CEC) [3], swelling property [4,5], and unique clay mineral microstructure [6], there are no reports on the effects of illite on crude oil biodegradation (Table 2), especially, biodegradation of heavy oil. Heavy oil is an important resource accounting for 21.3% of the global known recoverable oil resources [7]. As research on heavy oil biodegradation is difficult [8] due to the higher viscosity and density of heavy oil [9], there are only a few reports available on the influence of clay minerals on the biodegradation of heavy oil, except those affected by specific environmental conditions at a certain time (Table 2). 

Most of the previous studies had focused on unidentified microbial community easily degradable crude oil systems (Table 2), and only a few identified microbial species were involved, such as Alcanivorax borkumensis and Pseudomonas aeruginosa (Table 2). The macro phenomenon of crude oil degradation by an unidentified microbial community is a sum of selective degradations of crude oil by all microorganisms in the microbial community and varies with the composition and abundance of microbial species in the microbial community [10,11], which is an obstacle to correctly understand the role of clay minerals in crude oil biodegradation. As a crude oil degrading bacterium [12,13,14,15], *Pseudomonas stutzeri* has been detected in heavy oil reservoirs [16]; however, there are no studies on heavy oil biodegradation by this bacterium, not to mention the influence of clay minerals.

The present study is the first to investigate the interaction between illite and a *P. stutzeri*-heavy oil biodegradation complex (*Pst*HO). The biodegradation of heavy oil by *P. stutzeri* in the presence of illite was examined, and different effects of illite on the biodegradation of SHs and AHs compounds were determined. Furthermore, biogenic kaolinization of illite without smectite formation [17,18] in the illite-*Pst*HO was reported for the first time.

**Table 2 microorganisms-11-00330-t002:** Effect of clay minerals on microbial degradation of petroleum hydrocarbons.

No.	Clay Mineral	Clay Source/ Treatment Method	Degradation Duration ^1^	Substrate	Effect	Mechanism of Influence	Aerobic/Anaerobic	Degrader	Reference
1	Mix	Collected	56 d	Crude oil	Stimulation for SHs, Neutral for AHs	Increase biological accessibility	Aerobic	Microbial community	[19]
2	Kao	Purchased	24 d	Heavy oil in the environment	Stimulation	C-O-Na-Si stimulates metabolism	Aerobic	Microbial community	[20]
3	Mon	Collected	105 d	Heavy oil in the environment	Stimulation	Stimulate growth and buffer pH	Aerobic	*Pseudomonas* *aeruginosa* + Microbial community	[21]
	Kao								
4	Mon	Collected	36 mon	Heavy oil in the environment	Stimulation	Stimulate growth and buffer pH, C-O-Na-Si stimulates metabolism	Aerobic	Microbial community	[22]
	Mon				Stimulation				
	Kao-low				Stimulation				
	Kao-high				Overall inhibition, inhibition for SHs and AHs, stimulation for Rs and As	Low SSA and CEC			
5	Ver	Purchased	20 d	Naphthalene, Anthracene	Stimulation	Protect from toxicity	Aerobic	Microbial community	[23]
6	Mon	Purchased	60 d	Crude oil	Stimulation	Adsorbent	Aerobic + Anaerobic	Microbial community	[24]
	Mon-Org	Modified by DDDMA bromide			Neutral	Poor adsorption			
	Mon-Acid	HCl modified			Neutral	Poor adsorption			
	Mon-Na	NaCl modified			Stimulation	Adsorbent			
	Mon-K	KCl modified			Neutral	Poor adsorption			
	Mon-Ca	CaCl_2_ modified			Stimulation	Poor adsorption			
	Mon-Fe	FeCl_3_ modified			Stimulation	Poor adsorption			
7	Mon	Purchased	21 d	SHs in crude oil	Stimulation	High SSA	Aerobic + Anaerobic	Microbial community	[25]
	Mon-Acid	HCl modified			Inhibition	Low pH			
	Mon-Org	DDDMA bromide modified			Inhibition	Adsorption is blocked and local bridging effect are weakened			
	Pal	Collected			Stimulation	High SSA			
	Pal-Acid	HCl modified			Inhibition	Low pH			
	Sap	Collected			Neutral	/			
	Sap-Acid	HCl modified			Inhibition	Low pH			
	Sap-Org	DDDMA bromide modified			Inhibition	/			
	Kao	Purchased			Inhibition	No local bridging effect, Low SSA			
	Kao-Acid	HCl modified			Inhibition	Low pH			
8	Mon-Na	Purchased	60 d	Crude oil	Stimulation	High SSA and CEC	Aerobic + Anaerobic	Microbial community	[26]
	Mon-Org	Modified by DDDMA bromide			Inhibition	Hydrophobicity			
	Sap	Collected			Stimulation	High SSA and CEC			
	Sap-Org	Modified by DDDMA bromide			Neutral	Hydrophobicity			
9	Kao	Purchased	60 d	Crude oil	Inhibition	Low SSA and CEC	Aerobic + Anaerobic	Microbial community	[27]
	Pal	Collected			Stimulation	High SSA and CEC			
	Sap	Collected			Neutral	/			
	Mon	Purchased			Stimulation	High SSA and CEC			
	Kao-Acid	HCl modified			Inhibition	Reduce pH to form biological toxicity			
	Pal-Acid	HCl modified			Inhibition				
	Sap-Acid	HCl modified			Inhibition				
	Mon-Acid	HCl modified			Inhibition				
10	Mon	Purchased	60 d	Crude oil	Stimulation	/	Aerobic + Anaerobic	Microbial community	[28]
	Mon-Na	NaCl modified			Stimulation	SSA, CEC			
	Mon-K	KCl modified			Inhibition	Adsorbent			
	Mon-Mg	MgCl_2_ modified			/	/			
	Mon-Ca	CaCl_2_ modified			Stimulation	SSA, CEC			
	Mon-Zn	ZnCl_2_ modified			Inhibition	Adsorbent			
	Mon-Al	AlCl_3_ modified			Inhibition	Increase acidity			
	Mon-Cr	CrCl_3_ modified			Inhibition	Adsorb and increase acidity			
	Mon-Fe	FeCl_3_ modified			Stimulation	CEC			
11	Mon	Purchased	60 d	Phenanthrene and dibenzothiophene compounds	Stimulation	/	Aerobic + Anaerobic	Microbial community	[29]
	Mon-Acid	HCl modified			Inhibition				
	Mon-Org	Modified by DDDMA bromide			Stimulation				
	Mon-Na	NaCl modified			Stimulation				
	Mon-K	KCl modified			Inhibition				
	Mon-Ca	CaCl_2_ modified			Stimulation				
	Mon-Zn	ZnCl_2_ modified			Inhibition				
	Mon-Cr	CrCl_3_ modified			Inhibition				
	Mon-Fe	FeCl_3_ modified			Stimulation				
12	Mon-Na	NaCl modified	60 d	AHs in crude oil	Stimulation	SSA, CEC	Aerobic	Microbial community	[30]
	Mon-K	KCl modified			Inhibition	Adsorption and hydrophobic siloxane surface exposure			
	Mon-Mg	MgCl_2_ modified			Stimulation	High SSA and CEC, Local bridging effect			
	Mon-Ca	CaCl_2_ modified			Stimulation	SSA, CEC and local bridging effect			
	Mon-Zn	ZnCl_2_ modified			Inhibition	Adsorbed aromatic hydrocarbons			
	Mon-Al	AlCl_3_ modified			Inhibition	Low SSA			
	Mon-Cr	CrCl_3_ modified			Inhibition	Low SSA			
	Mon-Fe	FeCl_3_ modified			Stimulation	High SSA			
13	Bentonite-Surf + Acid	Surfactant and palmitic acid modified	21 d	Phenanthrene and cadmium contaminated soil	Stimulation	Adsorb cadmium to reduce toxicity	Aerobic	Microbial community	[31]
	Bentonite-Surf	Surfactant modified			Stimulation				
	Bentonite	Purchased			Stimulation				
14	Calcium bentonite	Collected	30 d/60 d	Crude oil in the environment	Stimulation	High SSA	Aerobic + Anaerobic	Microbial community	[32]
	Fuller soil	Collected			Stimulation				
	Kao	Collected			Stimulation				
	Eutrophic bentonite	Mixed with nutrients containing nitrogen, phosphorus and potassium			Stimulation	Fixed nutrients			
	Eutrophic fuller soil				Stimulation				
	Eutrophic kaolinite				Stimulation				
15	Pal	Purchased	5 d	Phenanthrene(C^14^)	Stimulation	Stimulate biofilm formation and accommodate extracellular enzymes	Aerobic	*Burkholderia sartisoli*	[33]
	Pal-Ther	Thermal modification			Stimulation	Reduced cooperation with the phenanthrene			
16	Mon	Collected	21 d	Phenanthrene(C^14^)	Stimulation	High SSA and CEC	Aerobic	*Burkholderia* *sartisoli* RP007 + Microbial community	[34]
	Mon-Acid	HCl modified			Stimulation	Element release, increase SSA and CEC			
	Mon-Alk	NaOH modified			Stimulation				
	Pal	Purchased			Stimulation	High SSA and CEC			
	Pal-Acid	HCl modified			Stimulation	Element release, increase SSA and CEC			
	Pal-Alk	NaOH modified			Stimulation				
17	Mon	Purchased	/	Crude oil	Stimulation	Stimulate contact with nutrients	Aerobic + Anaerobic	Microbial community	[35]
	Sap	Collected			Stimulation	Increase nutrient utilization			
	Mon-Org	Modified by didecyl dimethyl ammonium bromide			Inhibition for LMW AHs	Adsorbent			
	Sap-Org				Inhibition for LMW AHs and stimulation for phenanthrene	Adsorbent			
18	Mon	Purchased	21 d	AHs in crude oil	Stimulation	High SSA and CEC	Aerobic + Anaerobic	Microbial community	[36]
	Sap	Collected			Stimulation				
	Pal	Purchased			Stimulation	Channel structure			
	Kao	Purchased			Inhibition	Influence of impurities			
	Mon-Acid	HCl modified			Inhibition	Decrease pH			
	Sap-Acid	HCl modified			Inhibition				
	Pal-Acid	HCl modified			Inhibition				
	Kao-Acid	HCl modified			Inhibition				
19	Kao	Purchased	48 h	Phenanthrene	Stimulation	Silicon/oxygen atoms stimulate biological effects	Aerobic	*Sphingomonas* sp. GY2B	[37]
	Quartz	Purchased			Stimulation				
20	Nontronite	Collected	37 d	Crude oil	Stimulation	Stimulate ion exchange and nutrient absorption	Aerobic	*Alcanivorax borkumensis*	[38]
21	Bentonite	Purchased	70 d	AHs and cadmium contaminated soil	Stimulation	Adsorption of heavy metals	Aerobic + Anaerobic	Microbial community	[39]
	Bentonite- Surf	Modified by Arquad			Stimulation	Improve biological activity			
	Bentonite- Surf + Acid	Modified by Arquad and palmitic acid			Stimulation	Adsorb cadmium to reduce toxicity			
22	Pal	Collected	2 mon	Crude oil contaminated soil	Neutral	/	Aerobic	Microbial community	[40]
	Pal-Org	Modified by DDTMA bromide			Neutral				
23	Illite	Purchased	56 d	Heavy oil	Inhibition for all SHs and 50 AHs, stimulation for 45 AHs	Adsorption and cation-π	Aerobic	*Pseudomonas stutzeri*	This study

Note: Kao-low is low defect kaolinite; Kao-high is high defect kaolinite; Bentonite-Surf + Acid is surfactant and acid modified bentonite; Bentonite-Surf is surfactant modified bentonite; Pal-Ther is thermal modification palygorskite; Mon-Alk is alkali modified montmorillonite; Pal-Alk is alkali modified palygorskite; Pal-Org is organically modified palygorskite; DDDMA is didecyldimethylammonium; DDTMA is dodecyltrimethylammonium; LMW is low molecular weight. See Table 1 for the other abbreviations; ^1^. h stands for hour, d stands for day, mon stands for month; / Indicates that there is no relevant information in the literature.

## 2. Materials and Methods

### 2.1. Materials

#### 2.1.1. Illite 

Illite (Chengde, China) was purchased from Tianjin Guangfu Co. Ltd., Tianjin, China. According to Wang’s research [41], the composition of illite from Chengde is: 53.5% SiO_2_; 27.67% Al_2_O_3_; 1.14% Fe_2_O_3_; 0.036% FeO;1.25% MgO; 0.64% CaO; 0.75% Na_2_O; 7.77% K_2_O, and 5.25% loss-on-ignition (LOI). This natural illite clay is initially a wet mud cake without particle size screening, and its CEC is 140 meq/kg [4].

#### 2.1.2. Strain *Pseudomonas stutzeri* L1SHX-3X

Aerobic *P. stutzeri* strain L1SHX-3X was isolated from a heavy oil–water mixture of the production well in Liaohe Oil Field, China, and was identified [42,43] by the Research Centre for Geomicrobial Resources and Application, China University of Petroleum, Beijing, China. The strain was preserved by converting it into freeze-dried powder [44] and stored in a refrigerator (Thermo Fisher Scientific, Waltham, MA, USA) at −80 °C.

#### 2.1.3. Heavy Oil

The heavy oil L1YJC23 used in this study was collected from JC23 well in the Jin 45 Block of Liaohe Oil Field and preserved in an airtight plastic bucket [42].

All the chemical agents were of analytical grade and supplied by Tianjin Fuchen Chemical Reagents Factory, Tianjin, China.

### 2.2. Methods

#### 2.2.1. Experiment on the Interaction between Illite and *P. stutzeri*-Heavy Oil Complex

Illite was sifted through a 200-mesh sieve (74 μm) and dried in an oven (75 °C, 48 h), and its particle size and SSA were measured.

Reactivation of *P. stutzeri* was achieved by adding 1 mL of freeze-dried powder of *P. stutzeri* (obtained with 500 mL of *P. stutzeri* L1SHX-3X culture medium in logarithmic growth stage) to 4 mL of modified Van Niel culture medium [45] (MVN-R) under sterile conditions and incubating in a shaker (120 rpm, 35 °C) for 2 days [44]. The MVN-R contained 4.50 g/L C_6_H_12_O_6_, 0.10 g/L of NH_4_Cl, 0.04 g/L of MgCl_2_·6H_2_O, 0.05 g/L of KH_2_PO_4_, 0.50 g/L of Na_2_CO_3_, 0.10 g/L of Na_2_S·9H_2_O, and 0.10 g/L of NaCl. After 2 days of incubation, 4 mL of the culture were mixed with 40 mL MVN-R under the same culture conditions [44] until *P. stutzeri* concentration reached 10^8^ cell/mL under an Eclipse Ni-U upright microscope (Nikon, Tokyo, Japan) (Appendix A) for standby.

Before starting the experiment, the viscosity and density of heavy oil were measured, and the genes of in situ microorganisms in heavy oil were analyzed. In order not to affect the composition of the heavy oil, it was not sterilized.

The illite and MVN were first placed in 250 mL conical flasks (Table 3). The MVN was consistent with MVN-R except that it contained no C_6_H_12_O_6_. Then, the flasks were sealed with high-temperature resistant sealing films (BKMAN, Shanghai, China), and placed in a sterilizer (Zealway, Wilmington, DE, USA) for 20 min (121 °C, 1 atm). Then, *P. stutzeri* culture medium and heavy oil were added into the conical flasks under sterile conditions, and the flasks were sealed using Parafilm^®^ sealing film (Bemis, Neenah, WI, USA), which is germproof and breathable (150 cm^3^/m^2^/24 h at 22.78 °C for O_2_). All the flasks were incubated in a shaker (STIK, Shanghai, China) at 35 °C, 120 rpmfor 56 days.

A total of five groups of illite-*Pst*HO experiments were established (Table 3). The control groups, P0I0, and P0I8 were used to determine the effects of MVN and illite on heavy oil without *P. stutzeri* (Table 3). The experimental groups, P2I0, P2I8, and P2I32, contained the constant volume of *P. stutzeri* culture medium and different masses of illite (Table 3) to determine the effect of illite content. P2I8 and P2I32 contained 8 and 32 g of illite, respectively, simulating two environmental conditions with different solid contents (15.8% and 43.0%, respectively) and partly representing marine sedimentary and humid soil [22,39], respectively. Three parallel replicates were established for each group (Table 3) for error analysis. 

After 56 days, gas samples were collected from the conical flask by piercing the sealing film with a syringe and analyzed by gas chromatography (GC). The contents of the conical flask were filtered using 2-μm filter paper to obtain liquid and illite-heavy oil mixtures. The pH, conductivity (σ), and redox potential (Eh) of the liquid were determined, and the illite-heavy oil mixtures were mixed with 50 mL of organic solvent mixture (C_6_H_14_ and C_3_H_6_O at a ratio of 1:4) [46] and subjected to ultrasonication (60 min, 45 °C) to extract heavy oil. Further, heavy oil extraction was performed using solvent CH_2_Cl_2_ until CH_2_Cl_2_ leachate had no absorption in the spectral range of 200–400 nm on an ultraviolet-visible spectrophotometer (Varian Cary 100 UV-Vis, Agilent, Santa Clara, CA, USA) [47]. The heavy oil dissolved in an organic solvent mixture, and CH_2_Cl_2_ was collected and concentrated to 4 mL using a rotary evaporator (Yarong, Qingdao, China) at 65 °C, and then, further dried at room temperature (24 °C) to a constant weight [47] (weighed every 4 h, with differences of three consecutive measurements maintained within 0.0010 g) to obtain heavy oil [47]. Fractions analysis of SHs, AHs, resins (Rs), and asphaltenes (Aps) (SARA) in heavy oil [48] was performed, and the fractions of SHs and AHs were subjected to gas chromatography–mass spectrometry (GC–MS). Illite with CH_2_Cl_2_ was dried in an oven at 75 °C to a constant weight (as described previously) and further analyzed by X-ray diffraction (XRD) and scanning electron microscopy (SEM).

#### 2.2.2. Measurements of Particle Size and SSA of Illite 

To measure the particle size and SSA of illite, 2 g of illite was mixed with deionized water dispersant and examined under a blue light source with a 466 nm wavelength on a Mastersizer 3000 (Malvern Panalytical, Great Malvern, UK) [49]. The measurement was performed in triplicate, and the average value was determined.

#### 2.2.3. Measurements of Viscosity and Density of Heavy Oil

To measure the viscosity of heavy oil, a BS/U tube viscometer (Cannon, Huntington, NY, USA) loaded with heavy oil L1YJC23 was vertically placed into a constant-temperature water bath at 40 °C for 20 min. The time when the heavy oil reached the specific liquid level was recorded to obtain the viscosity [50].

To determine the density of heavy oil, the test temperature was set to 20 °C, and the heavy oil was injected into the U-shaped pipe of the digital density meter (DMA 4501, Anton Paar, Graz, Austria) using a syringe [51]. 

Measurements of viscosity and density of heavy oil were performed in triplicates individually, and the average values were obtained.

#### 2.2.4. Gene Sequence Analysis of In Situ Microorganisms in Heavy Oil

Biomass from heavy oil L1YJC23 was extracted with isooctane, and the deoxyribonucleic acid was extracted from the biomass using Fast DNA Spin Kit (MP Biomedical, Irvine, California, USA) [42]. The genes were sequenced on a MiSeq^TM^ System (Illumina, San Diego, CA, USA) and analyzed using Galaxy Platform [42,43,52] (https://galaxyproject.org/, accessed on 19 January 2022).

#### 2.2.5. Gas Chromatography Analysis

10.0 mL of gas samples from the illite-*Pst*HO flasks and atmosphere were employed for GC analysis (Agilent HP-6890A, Agilent, Santa Clara, CA, USA). The capillary column was HP-5 (30 m × 0.32 mm, 5% phenyl methyl siloxane), carrier gas was He (99.99%), power gas was N_2_ (99.99%), and temperatures of the oven, front sample cell, front detector, and back detector were 50 °C, 100 °C, 200 °C, and 200 °C, respectively [53]. 

The differences in CO_2_, O_2_, and N_2_ contents between the gas samples from the illite-*Pst*HO flasks and atmosphere (Δ*G*, %) were calculated using the following Equation (1):(1)$$\Delta G=G_{a}-G^{*}$$
where *G_a_* is the percentage content of CO_2_, O_2_, or N_2_ in gas from the illite-*Pst*HO (%) experiment and *G^*^* is the percentage content of CO_2_, O_2_, or N_2_ in the atmosphere (%).

#### 2.2.6. Measurements of pH, Conductivity, and Redox Potential

The pH, σ, and Eh of the MVN and illite-*Pst*HO were measured using three probes (Inlab Expert Pro pH, Inlab Redox, and Inlab 731, respectively) on a Mettler Toledo SevenMulti™ (Mettler Toledo, Columbus, OH, USA) [53].

#### 2.2.7. Fractions Analysis of Heavy Oil

SARA analysis separates heavy oil components according to their polarizability and polarity by column chromatography [54]. In the present study, 30 mL n-hexane was added to 20.0–50.0 mg of heavy oil, and the Aps were filtered out with absorbent cotton after ultrasonication (5 min). The SHs, AHs, and Rs were obtained by using a chromatographic column (4 g of silica gel and 3 g of activated alumina) with n-hexane, dichloromethane, and ethanol [47,55].

The fraction content (*FC*, %) of SARA was calculated using the following Equation (2):(2)$$FC=\frac{M_{e}}{M}\times 100\%$$
where *M_e_* is the mass of each SARA in heavy oil L1YJC23 or illite-*Pst*HO (g) and *M* is the mass of heavy oil L1YJC23 used in each group (0.5 g).

#### 2.2.8. Gas Chromatography–Mass Spectrometry Analysis of Saturated and Aromatic Hydrocarbons

The compounds in the SHs and AHs of the heavy oil were characterized and quantified by GC–MS. Deuterated tetracosane (D_50_-*n*C_24_, 10 μg) and deuterated dibenzothiophene (D-substituted dibenzthiophene, 10 μg) were used as internal standards for SHs and AHs, respectively. Trace-DSQ mass spectrometer (Thermo Finnigan, San Jose, CA, USA) coupled to an HP 6890 gas chromatograph (Agilent, Santa Clara, CA, USA) was used for GC–MS analysis. The column was HP-5MS (30 m × 0.25 mm, ID) with a 0.25-μm coating, and He (99.99%) was used as the carrier gas. The oven temperature of the gas chromatograph was initially set to 50 °C and was, subsequently, increased to 120 °C at a rate of 20 °C/min, 250 °C at a rate of 4 °C/min, and 310 °C at a rate of 3 °C/min and maintained for 30 min. The mass spectrometer was operated in full-scan electron impact mode with an electron energy of 70 eV [56].

The residual mass content (*RMC*) of the SHs and AHs in the heavy oil L1YJC23 or illite-*Pst*HO was obtained by GC–MS. The *RMC* (μg/g), the degradation rate of heavy oil by *P. stutzeri* (*DR*, %), and the influence degree (*IND*, %) of illite on biodegradation were calculated using the following Equations (3)–(5):(3)$$RMC=k\frac{S_{a}m^{*}}{S^{*}m_{a}}\;$$
(4)$$DR=\frac{M^{*}-M_{p}}{M^{*}}\times 100\%$$
(5)$$IND=(\frac{M^{*}-M_{ip}}{M^{*}}-\frac{M^{*}-M_{op}}{M^{*}})\times 100\%$$
where *k* is the response coefficient, *S_a_* is the peak area of each compound in SHs and AHs, *S^*^* is the peak area of the internal standard, *m^*^* is the mass of the internal standard (10 μg), *m_a_* is the mass of heavy oil used in fractionation (g), *M^*^* is the *RMC* of each compound of SHs or AHs in heavy oil L1YJC23 (μg/g), *M_p_* is the *RMC* of each compound of SHs or AHs in P2I0, P2I8, or P2I32 with *P. stutzeri* (μg/g), *M_ip_* is the *RMC* of each compound of SHs or AHs in P2I8 and P2I32 with illite clay and *P. stutzeri* (μg/g), and *M_op_* is the *RMC* of each compound in SHs or AHs in P2I0 without illite and with *P. stutzeri* (μg/g). To facilitate comparison with previous studies, the *IND* of other clay minerals on biodegradation was calculated based on Equation (5).

In order to support the scientific and responsible biochemical processes [57] in the illite-*Pst*HO, mass balance and stoichiometry were performed in the data-checking process using the following Equations (6) and (7):(6)$$M\cdot FC_{SHs}=\frac{m_{a}}{k}\cdot \sum_{i=1}^{64}RMC_{SH}\cdot 10^{-6}$$
(7)$$M\cdot FC_{AHs}=\frac{m_{a}}{k}\cdot \sum_{i=1}^{96}RMC_{AH}\cdot 10^{-6}$$
where *FC_SHs_* is the fraction content of SHs (%), *RMC_SH_* is the *RMC* of each compound in SHs (μg/g), *FC_AHs_* is the fraction content of AHs (%), and *RMC_AH_* is the *RMC* of each compound in AHs (μg/g).

#### 2.2.9. X-ray Diffraction Analysis of Illite

XRD analysis of illite was performed using D8 Advance X-ray diffractometer (Bruker, Billerica, MA, USA). The tube voltage was 40 kV and the tube current was 25 mA [53].

#### 2.2.10. Scanning Electron Microscopy Analysis of Illite

The illite was sprayed with gold and observed using TESCAN VEGA 3 (Czech, TESCAN, Brno, Czech Republic) SEM with an electron detector at an accelerating voltage of 20 kV [53].

## 3. Results and Discussion

### 3.1. Effect of Illite on Heavy Oil and In Situ Microorganisms in Heavy Oil

Illite was incapable of altering the SARA fractions of heavy oil in the absence of *P. stutzeri*; however, it modified the existing state of heavy oil. In this study, the recovery efficiency of eluted SARA fractions ranged from 90% to 99%, completely meeting the requirements for its effectiveness (85–115%) [58]. The SARA fractions of P0I0, P0I8, and heavy oil L1YJC23 were consistent (Figure 1). Due to its high viscosity (1967 MPa·s) and density (0.949 g/cm^3^), heavy oil remained suspended and dispersed in the MVN as droplets of different sizes or adhered to the inner wall of the conical flask in the absence of illite. However, the adsorption of illite particles (8.67 μm; Appendix A) caused the aggregation of oil droplets and the formation of larger illite-heavy oil mixtures. The mixtures were mostly covered with illite due to the relative excess proportion of illite (8/32 g) to heavy oil (0.5 g).

Illite did not affect the activity of in situ microorganisms in heavy oil. The gene sequence analysis indicated that the in situ microorganisms in heavy oil L1YJC23 included 61.1% aerobic and 14.1% facultative anaerobic microorganisms (anaerobic and unidentified microorganisms accounted for 17.1% and 7.7%, respectively) (Appendix A). Among the in situ microorganisms in heavy oil, Pseudomonas was the dominant genus with the largest number of reads (20.1%, Appendix A). Aerobic and facultative anaerobic microorganisms (Appendix A) activated by MVN consumed O_2_ and produced CO_2_, which resulted in gas content differences of CO_2_ and O_2_ between groups of P0I0 and P0I8 without *P. stutzeri* and atmosphere (Appendix A). Comparison of P0I0 with P0I8 revealed that the addition of 8 g of illite did not cause any changes in the CO_2_ and O_2_ contents (Appendix A), indicating that illite did not affect the activity of in situ microorganisms in heavy oil L1YJC23. Moreover, in situ microorganisms did not alter the SARA fractions of heavy oil, irrespective of the presence or absence of illite (Figure 1). 

### 3.2. Illite Effect on P. stutzeri-Heavy Oil Complex

#### 3.2.1. Illite Effect on Activity of *P. stutzeri*

Illite slightly inhibited the activity of *P. stutzeri*. In the absence of illite, P2I0 presented the highest CO_2_ and lowest O_2_ contents (Appendix A) due to the metabolism of the aerobic bacterium *P. stutzeri* [12]. The pH, σ, and Eh of P2I0 were the lowest (Appendix A) because of the consumption of inorganic salts, O_2_, and petroleum hydrocarbons by *P. stutzeri* to produce CO_2_ and acidic compounds [12]. Consequently, P2I0 also showed the highest *DR* (23.0%) for heavy oil, followed by P2I8 (17.9%) and P2I32 (13.2%) (Figure 1). Comparison of P2I0, P2I8, and P2I32 revealed that the presence of illite reduced the CO_2_ content, increased the O_2_ content (Appendix A), reduced the changes in pH, σ, and Eh (Appendix A), and decreased *DR* (Figure 1). These results proved that illite had a negative effect on the activity of *P. stutzeri*, and the degree of the effect was positively related to the illite content (Figure 1 and Appendix A). However, the differences in the parameters (O_2_ content < 1%, CO_2_ content < 0.19%, pH < 1.5%, σ < 5.8%, Eh < 13.5%, and SHs fraction content < 9.8%; Figure 1 and Appendix A) were not sufficiently large to confirm any effect of illite on the metabolic pathways of *P. stutzeri*.

The slight inhibitory effect of illite on the activity of *P. stutzeri* might have been caused by a decrease in the bioaccessibility of heavy oil due to adsorption onto illite [59]. Illite was more likely to adsorb heavy oil droplets [60] than *P. stutzeri* [61] because both illite and *P. stutzeri* have negatively charged surfaces [12,22], and the excess illite coated the surface of the heavy oil and formed a barrier against *P. stutzeri*. In contrast, the moderate SSA (1.294 m²/g) and CEC [3] (140 meq/kg [4]) of illite were insufficient to elicit a positive effect on microbial activity similar to that of Kao with low SSA and CEC [22,25,27,36].

#### 3.2.2. Illite Inhibition of Saturated Hydrocarbons Biodegradation

The GC–MS results revealed that illite inhibited the biodegradation of all 64 SHs in heavy oil L1YJC23 (Appendix A). Hopane was degraded without the formation of 25-norhopane (Appendix A), indicating that the biodegradation level of heavy oil L1YJC23 was 7–8 on the Peters and Moldowan (PM) scale [8]. Furthermore, all n-alkane, alkyl cyclohexane, and isoprenoid components which should have been in the SHs of crude oil, were missing (Appendix A). The *RMC* of all the 64 SHs showed a trend of P2I0 < P2I8 < P2I32 (Appendix A), indicating that the *IND* of illite was positively correlated with its content. In SHs with content >70 μg/g, the *DR* of *P. stutzeri* without illite were related to the molecular weight (Figure 2). A higher molecular weight usually denotes stronger biodegradation resistance [62,63], thus leading to lower *DR*. The inhibition of illite on the biodegradation of SHs in heavy oil can be a positive protective way during microbial-enhanced oil recovery processes [53].

The adsorption of illite on SHs reduced their bioaccessibility, which was the main reason for the negative effect of illite. However, with an increase in the molecular weight of SHs, the adsorption of illite on SHs weakened, and the degree of reduction in bioaccessibility decreased, which led to a decrease in the inhibitory effect of illite (Figure 2). Generally, clay minerals with high SSA and CEC, such as Mon, Sap, and Pal, can stimulate biodegradation of SHs [25] (Appendix A); however, Kao and illite do not possess these properties, and hence, exert a negative effect on SHs biodegradation (Appendix A). Furthermore, the interlayer of illite does not contain divalent cations [60], and the positive effect of the local bridging effect [22,24,25,26,27,28,29,30,61] formed by the low concentration of divalent cations (Mg^2+^, 7.87 × 10^−6^ mol) provided by MVN was weak, which was not sufficient to alter the inhibition caused by adsorption.

#### 3.2.3. Two Effects of Illite on Aromatic Hydrocarbons Biodegradation

A total of 96 AHs were detected in the heavy oil L1YJC23 by GC–MS (Appendix A, Figure 3 and Appendix A), and were classified into five categories for ease of discussion, namely, naphthalene/phenanthrene/fluorene/biphenyl series and high-ring number (≥4) aromatic hydrocarbons (HRAHs) (Appendix A). The degradation rate of AHs by *P. stutzeri* was not affected by illite (Figure 1); however, the GC–MS results of AHs suggested that illite might have diverse effects on different AHs compounds (Appendix A). After the verification of fractions and GC–MS data in mass balance and stoichiometry (Equations (6) and (7)), we found that illite inhibited the biodegradation of 50 AHs, stimulated the biodegradation of 45 AHs, and had no obvious effect on biphenyl (Bph, see Appendix A for the abbreviations of AHs used in this study) (Appendix A).

The effect of illite on the biodegradation of AHs in *Pst*HO was affected by the number of aromatic rings, and illite appeared to inhibit the biodegradation of AHs with a high number of aromatic rings. For example, although both trimethyl naphthalene (TMN) and trimethyl phenanthrene (TMP) have three methyl groups (Figure 3), illite stimulated TMN biodegradation (Appendix A) but inhibited TMP biodegradation (Appendix A), which may have been caused by the addition of an aromatic ring. Furthermore, monomethyl biphenyl (MeBph), monomethyl pyrene (MePyr), and monomethyl chrysene (MeChr) have one methyl substituent (Figure 3); illite also stimulated MeBph biodegradation (Appendix A) with two aromatic rings and inhibited biodegradation of MePyr and MeChr (Appendix A) with four aromatic rings. The more fused the rings of AHs, the stronger the ability of the compounds to resist biodegradation [64], with illite being more inclined to inhibit their biodegradation.

The effect of illite on the biodegradation of AHs in *Pst*HO was affected by the number of methyl substituents. In the two-ring naphthalene series (Figure 3), as the number of methyl substituents increased from 0 to 5, the effect of illite varied from inhibition (0, 1, 2) to stimulation (3, 4) and then, to inhibition (5) (Appendix A). In general, with an increase in the number of methyl substituents, the steric hindrance, stability, and biodegradation resistance of naphthalene series compounds increase [64,65,66,67]. However, the effect of illite on the biodegradation of naphthalene series compounds was not consistent with this rule, and this phenomenon of altered influence with the increase in the methyl substitution number of naphthalene series was also exhibited by Mon [36] (Appendix A). The number of methyl substituents had a secondary effect when compared with the number of aromatic rings. For example, in the three-ring phenanthrene series (Figure 3), the number of methyl substituents increased from 0 to 3, and the effect of illite on their biodegradation was consistently negative (Appendix A), which was different from that on the naphthalene series (Appendix A). In contrast, although the effect of Mon was positive, consistency in the biodegradation of different compounds in the phenanthrene series was also observed [29,30] (Appendix A).

Furthermore, the influence of illite on the biodegradation of AHs in *Pst*HO was affected by the connection mode of the aromatic rings. For instance, although both naphthalene series and biphenyl series have two aromatic rings (Figure 3), illite inhibited the biodegradation of monomethyl naphthalene and dimethyl naphthalene (Appendix A) but stimulated MeBph and dimethyl biphenyl biodegradation (Appendix A). The key difference between these two series is that the naphthalene series is formed by the fusion of two benzene rings, whereas the biphenyl series is formed by linking two phenyl groups through a single covalent bond (Figure 3).

The impact of illite on the biodegradation of AHs in *Pst*HO was affected by heteroatoms. In the fluorene series, fluorene (Fle), dibenzothiophene (DBT), and dibenzofuran (DBF) have the same structure, except that the atom at position 5 is carbon, sulfur, and oxygen, respectively (Figure 3). Illite had different effects on the biodegradation of these compounds, and it is inhibiting Fle biodegradation and stimulating DBT and DBF biodegradation (Figure 3 and Appendix A). Similarly, this trend was also noted in the monomethyl substituents of these compounds, monomethyl fluorene, monomethyl dibenzothiophene, and monomethyl dibenzofuran (Figure 3 and Appendix A). However, DBT exhibited the strongest biodegradation resistance, followed by Fle and DBF [64,68]. Thus, the influence of illite on AHs biodegradation was inconsistent with the biodegradation resistance of AHs. Larger sulfur and oxygen atoms, when compared with carbon atoms, can alter the polarity of the molecules, which might be the reason for the different effects of illite on these compounds. Illite had varied effects on compounds containing different heteroatoms in the fluorene series, unlike other clay minerals (modified or not) (Appendix A). Kao and Sap could inhibit the biodegradation of Fle and DBT (and their methyl or ethyl substituent), whereas Mon and Pal could stimulate their biodegradation [29,30,36] (Appendix A).

The influence of illite on the biodegradation of AHs in *Pst*HO was affected by the carbon chain length in the substituent. HRAHs have more than four rings (Figure 3; 72–96), and illite inhibited the biodegradation of HRAHs with no substituent or only one methyl substituent (Figure 3, 72–90; Appendix A), but stimulated the biodegradation of triaromatic steroid (TAS) (Figure 3; 91–96; Appendix A) with the longer carbon chain substituents (C2, C3, C8, C9, and C10). It must be noted that it is common for unmodified clay minerals, such as Mon, Kao, Sap, and Pal, to stimulate the biodegradation of TAS [30,36] (Appendix A). 

The effect of illite on the biodegradation of AHs in *Pst*HO was not dominated by inhibition, as in the case of SHs. The positive effects of moderate SSA (and CEC) and weak local bridging effect were not sufficient to counteract the negative effects caused by adsorption, thus suggesting the possible occurrence of other positive mechanisms to explain the selective stimulation of illite to 45 AHs. Considering the facts that illite could not provide divalent cations (to form the local bridging effect) and this positive effect only occurred in AHs with π bonds, we speculated that the selective stimulation of illite was due to the cation-π [69] preferentially formed by monovalent cations and AHs with above five structure characteristics. 

### 3.3. Kaolinization of Illite in P. stutzeri-Heavy Oil Complex

This study is the first to report the kaolinization of illite without smectite formation in illite-*Pst*HO. The XRD patterns of illite clay in P0I8 and unused samples were the same as the standard pattern (Figure 4a, ICDD PDF 26-0911 Illite), indicating that MVN and heavy oil L1YJC23 (including in situ microorganisms therein) did not change the crystal characteristics of illite in the absence of *P. stutzeri* (Figure 4a). Although the main characteristic peaks (I_x_/I_max_ ≥ 0.5) [70] of illite crystals still remained in the intermediate (formed in the process of kaolinization of illite) of P2I8, they were significantly weakened (e.g., (002), (004), (006), and (136)) (Figure 4a). Other illite characteristic peaks (I_x_/I_max_ < 0.5) [70] of the intermediate decreased ((025), (115), and (−116)) and disappeared ((−113), (023), (−114), and (114)) (Figure 4a). At the same time, the intermediate of P2I8 exhibited new characteristic peaks of Kao, such as (001), (002), (003), and (−113) (Figure 4a). Furthermore, the edges of the illite particles in P0I8 were sharp (Figure 4b), while those of the intermediate particles of P2I8 were round and had attached precipitates of 250–1000 nm (Figure 4c). This phenomenon of rounded edges (Figure 4c) caused by local dissolution is similar to the kaolinization of illite under abiotic condition [5]. After the destruction of the edges of the illite crystals, K^+^ in the interlayers [71] was released to form cation-π [69], which stimulated the biodegradation of specific AHs. The organic products of biodegradation and ligands promoted the transformation of dissolved illite into the granular precipitate-aluminosilicate gel, which is considered the crucial first step in the two-step biological kaolinization [72]. The gel formed was deposited near the edges of the illite particles, which is considered to provide precursor materials and a crystallization environment for Kao. In the second step, biological metabolic activities further changed the local surrounding environment (pH, σ, and Eh) in and around the gel, resulting in the dissolution of the gel or rearrangement of its solid state to form Kao crystals [72]. Although previous studies have detected smectite (under the action of *Pseudogulbenkiania* sp.) in the process of kaolinization of illite [17,18], characteristic peaks of smectite were not observed in the present study, which may be attributed to the different microorganisms employed and the presence of heavy oil.

## 4. Conclusions

To the best of our knowledge, this is the first report on the interaction of illite with the *Pst*HO complex. Although illite clay exerted a negative effect on the total heavy oil degradation ratio, it stimulated the biodegradation of 45 AHs in heavy oil, and its kaolinization in the illite-*Pst*HO was observed for the first time. The selective inhibition/stimulation effects of illite clay on biodegradation, as a supplementary mechanism of quasi-sequential and/or selective degradation, can be applied to protect high-quality SHs during microbial enhanced oil recovery process and remediation of specific AHs pollutant in polluted environments. Kaolinization of illite clay could be further explored to explain clay minerals transformations in oil reservoirs. For future study, new multivariate complex systems, such as illite-*Pst*HO could lead to richer discoveries in the field.

## Figures and Tables

**Figure 1 microorganisms-11-00330-f001:**
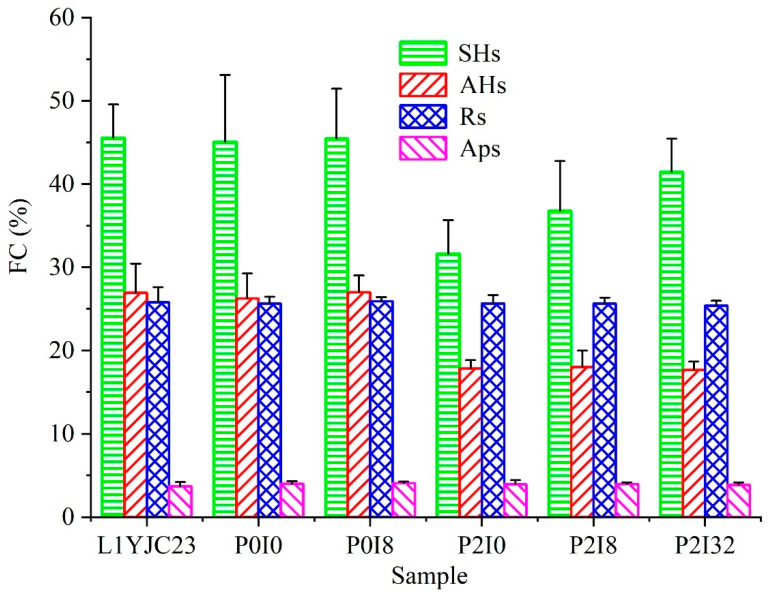
SARA analysis of heavy oil L1YJC23 and illite-*Pst*HO. Note: FC is fraction content; SHs, AHs, Rs, and Aps are saturated hydrocarbons, aromatic hydrocarbons, resins, and asphaltenes, respectively; see Table 3 for P0I0, P0I8, P2I0, P2I8, and P2I32.

**Figure 2 microorganisms-11-00330-f002:**
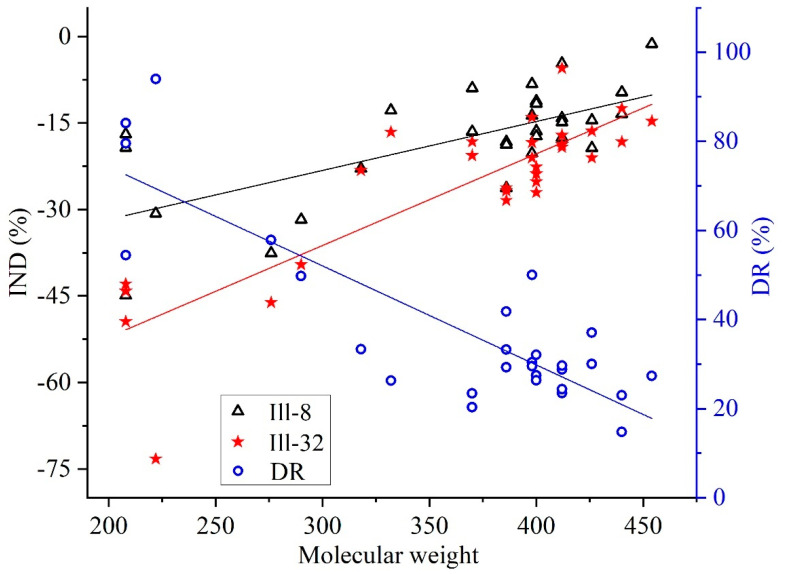
Relationship among the *IND* of illite, *DR* of *P. stutzeri* without illite, and molecular weight of SHs (content > 70 μg/g) in the illite-*Pst*HO. Note: *IND* is influence degree; *DR* is degradation rate; Ill-8 and Ill-32 represent 8 g and 32 g illite, respectively.

**Figure 3 microorganisms-11-00330-f003:**
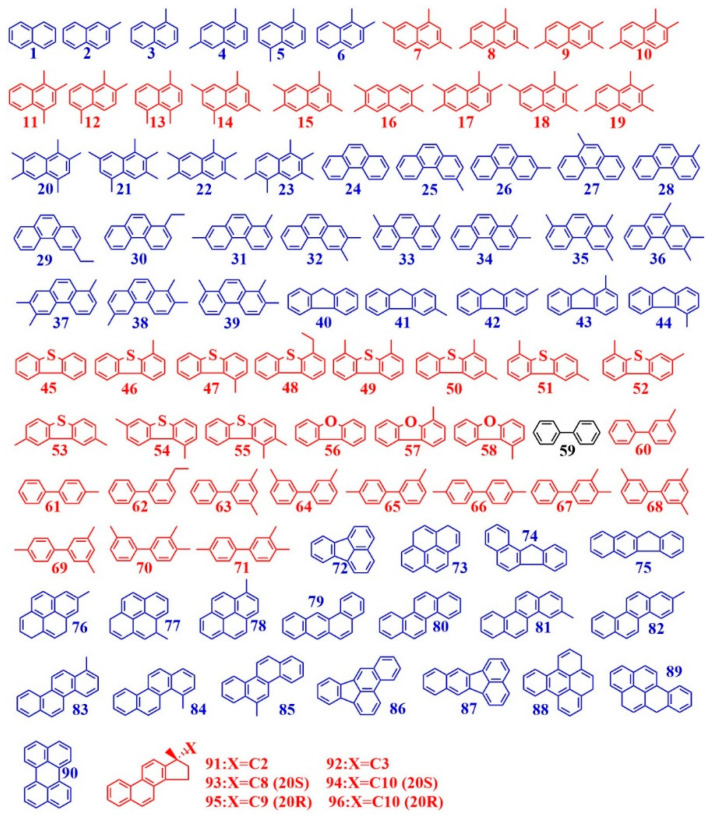
Molecular structures of AHs in heavy oil L1YJC23. Note: See Appendix A for No. of the AHs; blue, red, and black represent inhibition, stimulation, and uncertain effects of illite, respectively.

**Figure 4 microorganisms-11-00330-f004:**
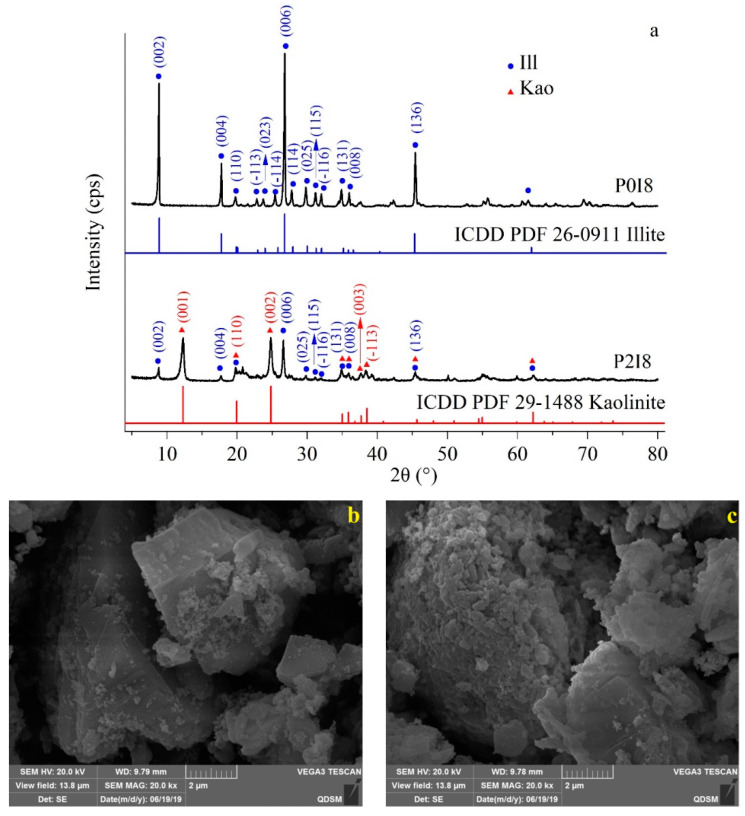
XRD (**a**) and SEM images (**b,c**) ((**b**) P0I8 and (**c**) P2I8) of the kaolinization in illite-*Pst*HO. Note: See Table 3 for P0I8 and P2I8; ICDD PDF 26-0911 and ICDD PDF 29-1488 represent the numbers of Powder Diffraction File, data source: International Centre for Diffraction Data (https://www.icdd.com, accessed on 12 October 2022).

**Table 1 microorganisms-11-00330-t001:** Abbreviation list for this paper.

No.	First Appearance Location	Content	Abbreviation	Occurrence Number
1	Introduction	montmorillonite	Mon	8
2		kaolinite	Kao	12
3		saponite	Sap	5
4		palygorskite	Pal	5
5		specific surface area	SSA	10
6		cation exchange capacity	CEC	9
7		*Pseudomonas stutzeri*-heavy oil system	*Pst*HO	25
8		saturated hydrocarbons	SHs	35
9		aromatic hydrocarbons	AHs	46
10	Materials and methods	modified Van Niel culture medium	MVN	16
11		gas chromatography	GC	5
12		conductivity	σ	9
13		redox potential	Eh	9
14		resins	Rs	4
15		asphaltenes	Aps	4
16		saturated hydrocarbons, aromatic hydrocarbons, resins, and asphaltenes	SARA	6
17		gas chromatography–mass spectrometry	GC–MS	8
18		X-ray diffraction	XRD	3
19		scanning electron microscope	SEM	4
20		fraction content	FC	3
21		the content difference of CO_2_, O_2_ and N_2_ between illite-*Pst*HO and atmosphere	ΔG	3
22		residual mass content	*RMC*	11
23		degradation rate	*DR*	4
24		influence degree	*IND*	9
25	Results and discussion	Peters and Moldowan	PM	2
26		high ring number (≥4) aromatic hydrocarbons	HRAHs	7
27	Appendix A	8 g illite	Ill-8	29
28		32 g illite	Ill-32	29
29		organic modified montmorillonite	Mon-Org	18
30		acid modified montmorillonite	Mon-Acid	29
31		sodium ion modified montmorillonite	Mon-Na	10
32		potassium ion modified montmorillonite	Mon-K	21
33		calcium ion modified montmorillonite	Mon-Ca	21
34		iron ion modified montmorillonite	Mon-Fe	21
35		acid modified palygorskite	Pal-Acid	10
36		acid modified saponite	Sap-Acid	8
37		organic modified saponite	Sap-Org	3
38		acid modified kaolinite	Kao-Acid	8
39		magnesium ion modified montmorillonite	Mon-Mg	8
40		zinc ion modified montmorillonite	Mon-Zn	9
41		aluminum ion modified montmorillonite	Mon-Al	8
42		chromium ion modified montmorillonite	Mon-Cr	9

Note: In order of appearance in the context.

**Table 3 microorganisms-11-00330-t003:** Experimental setup of the illite-*Pst*HO.

Group ^#^	Volume of *P. stutzeri* Culture (mL)	Optical Density of Culture (Cell/mL)	Mass of Illite (g)	Mass of Heavy Oil (g)	Volume of MVN (mL)	Duration (d)	Number of Parallel Replicates
P0I0	0	10^8^	0	0.50	40.0	56	3
P0I8	0		8.00				
P2I0	2.0		0.00				
P2I8	2.0		8.00				
P2I32	2.0		32.00				

Note: See Table 1 for *Pst*HO and MVN. ^#^. P represents *P. stutzeri*, followed by 0 or 2, which denotes the volume of *P. stutzeri* culture; I indicates illite, followed by 0, 8, or 32, which shows the mass of illite.

## Data Availability

Data will be made available on request.

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
