# Peer review of "Interaction between Illite and a Pseudomonas stutzeri-Heavy Oil Biodegradation Complex"

_microorganisms, 2023, doi:10.3390/microorganisms11020330_

Round 1

Reviewer 1 Report

Pollution of the environment by crude oil and oil products will be of high interest for a long time. Therefore, in-depth studies of biodegradation and interaction of oil-degrading bacteria with the environment are undoubtedly important.

Minor comments:

- The introduction provides a sufficient background for understanding the problem. But I noticed that Table summarizes references published up to 2018. Have there been no studies in the past 4 years? Why is that?

- Line 63: I propose to include a description and physical properties of illite in the section 2.1.1.

- Table 2: How did you choose the optimal volume of biomass (2 mL) to be applied?

- Table S1: Searching for the necessary abbreviation in Supplementary materials is not convenient for the reader. I suggest moving the list of abbreviations to the manuscript. I also recommend deciphering abbreviations in figure captions.

Reviewer 2 Report

Comments are given in the paper. It is necessary to refine the experimental part. The authors do not explain the experimental setup or culture adaptation, which is necessary.

Round 2

Reviewer 2 Report

The paper can be accepted in this form.
